# Dynamics of Fanconi anemia protein D2 in association with nuclear lipid droplet formation

Tomoya Hotani[1,2], Motonari Goto[1,2], Yukie Otsuki[1,2], Shun Matsuda[3,*], Nobuhiro Wada[4], Masakazu Shinohara[5,6], Tomonari Matsuda[3], Masayuki Yokoi[1,2], Kaoru Sugasawa[1,2], Yuki Ohsaki[4,‡] and Wataru Sakai[1,2,‡]

## ABSTRACT

Fanconi anemia is a rare genetic disease caused by the loss of function of one of the 23 associated genes and is characterized by bone marrow failure, cancer predisposition and developmental defects. The proteins encoded by these genes (FANC proteins) mainly function in DNA damage response and repair. Although FANC deficiency has multiple effects on the regulation of lipid metabolism, the molecular function of FANC proteins in the context of Fanconi anemia pathology remains unclear. In the present study, we demonstrate that FANCD2, a key component of FANC proteins, interacts with factors involved in fatty acid biosynthesis or sphingolipid metabolism and that FANCD2 deficiency downregulates the cellular levels of fatty acids. Moreover, a portion of FANCD2 is localized to nuclear lipid droplets in response to oleic acid (OA) treatment. These subcellular dynamics are independent of FANCD2 monoubiquitylation, which is essential for the DNA damage response. Collectively, these findings demonstrate that FANCD2 responds to not only DNA damage but also OA exposure, providing insights into the pathogenesis of lipid dysregulation in Fanconi anemia.

KEY WORDS: Lipid droplet, Lipid metabolism, DNA damage response, Fanconi anemia

## INTRODUCTION

Fanconi anemia is a rare genetic disease in humans characterized by bone marrow failure, cancer predisposition and developmental defects (Auerbach, 2009). To date, more than 20 genes have been identified to be responsible for Fanconi anemia (Homan et al., 2025; Niraj et al., 2019). The products of these genes (FANC proteins) are mainly involved in the maintenance of genome stability, particularly in DNA interstrand crosslink repair, which is known as the Fanconi anemia pathway. Ten FANC proteins and other FANC-related factors form a Fanconi anemia core complex with E3 ubiquitin ligase activity, which functions as an upstream regulator of the Fanconi anemia pathway. In response to DNA damage, FANCD2 and FANCI form a heterodimer – known as the ID complex – and are monoubiquitylated by the Fanconi anemia core complex (Smogorzewska et al., 2007). This reaction is a crucial step in orchestrating the entire Fanconi anemia pathway and the downstream repair process (Ishiai et al., 2017). Over recent decades, FANC proteins and related factors have been isolated, revealing the molecular mechanisms underlying the Fanconi anemia pathway (Ceccaldi et al., 2016; Sakai and Sugasawa, 2019). Recent studies have shown that endogenous aldehydes have a significant effect on Fanconi anemia symptoms (Garaycoechea et al., 2012; Langevin et al., 2011; Pontel et al., 2015). Defects in FANC proteins result in the dysregulation of aldehyde-induced DNA damage response and repair, eventually leading to Fanconi anemia symptoms, such as bone marrow failure and cancer predisposition. Moreover, endocrine and metabolic abnormalities are reported as one of the common phenotypes of Fanconi anemia (Giri et al., 2007; Petryk et al., 2015). Approximately 80% of individuals with Fanconi anemia have at least one endocrine abnormality. More than half of the individuals with Fanconi anemia have impaired lipid metabolism. Consistent with these Fanconi anemia symptoms, alterations in lipid metabolism resulting from FANC deficiency have been reported at the cellular level. Mesenchymal stromal cells from *Fanca* and *Fancd2* knockout mice exhibit abnormal lipid profiles, indicating disturbances in fatty acid and lipid metabolism (Amarachintha et al., 2015), and *Fancd2*-deficient male mice exhibit altered hepatic lipid and bile acid metabolism when fed the Paigen diet (Moore et al., 2019). FANC deficiency has been reported to affect glycosphingolipid metabolism (Zhao et al., 2018). Additionally, FANC-deficient human cells show accumulation of lipid droplets (LDs) (Ravera et al., 2019), which constitute an organelle crucial for lipid metabolism. LDs are composed of lipid monolayers in which neutral lipids, triacylglycerol and cholesterol ester are stored. Various factors related to lipid metabolism have been identified in the LD membrane (Bersuker et al., 2018), with these factors playing an important role in lipid metabolism, such as energy production, membrane trafficking between organelles and lipid accumulation (Olzmann and Carvalho, 2019). Although LDs are classically defined as cytoplasmic organelles, it has recently been reported that LD formation also occurs in the nucleus in various cell types and tissues (Layerenza et al., 2013; McPhee et al., 2024; Ohsaki et al., 2016; Romanauska and Köhler, 2018, 2021; Uzbekov and Roingeard, 2013; Wang et al., 2013; Zadoorian et al., 2023), suggesting that lipid metabolism via nuclear LDs (nLDs) occurs even in the nucleus.

[1]Biosignal Research Center, Kobe University, Kobe 657-8501, Japan. [2]Department of Biology, Graduate School of Science, Kobe University, Kobe 657-8501, Japan. [3]Research Center for Environmental Quality Management, Graduate School of Engineering, Kyoto University, Otsu 520-0811, Japan. [4]Division of Cell and Tissue Morphology, Department of Anatomy, School of Medicine, Sapporo Medical University, Sapporo 060-8556, Japan. [5]Division of Molecular Epidemiology, Kobe University Graduate School of Medicine, Kobe 650-0017, Japan. [6]The Integrated Center for Mass Spectrometry, Kobe University Graduate School of Medicine, Kobe 650-0017, Japan.
*Present address: Department of Environmental Engineering, Graduate School of Engineering, Kyoto University, Kyoto 615-8530, Japan.

‡Authors for correspondence (yohsaki@sapmed.ac.jp; wsakai@phoenix.kobe-u.ac.jp)

M.Y., 0000-0001-9727-0258; K.S., 0000-0001-7937-4053; W.S., 0000-0002-6561-5773

*Journal of Cell Science*

Increasing evidence suggests a functional link between FANC proteins and lipid metabolism. However, the intracellular dynamics of FANC proteins in relation to lipid metabolism remain unclear. In the present study, we undertook gene ontology analysis following screening for FANCD2-binding factors revealing that FANCD2 interacts with factors related to fatty acid and lipid metabolism. Furthermore, we find that FANCD2 localizes around nLDs in response to OA exposure. These results suggest that FANCD2 responds not only to DNA damage but also to nLD formation.

## RESULTS

### Lipid metabolism-related factors as FANCD2-interacting proteins

To obtain novel insights into FANCD2 function independent of the DNA damage response, we performed mass spectrometry using the human osteosarcoma U2OS cell line stably expressing FLAG epitope-tagged FANCD2 (FLAG–FANCD2) (Fig. S1A). The cell lysate for mass spectrometry analysis was prepared from the cell culture without treatment with an exogenous DNA-damaging agent. The protein complex with FLAG–FANCD2 was subjected to SDS-PAGE followed by Coomassie Brilliant Blue (CBB) staining (Fig. S1B) and mass spectrometric analysis as described in the Materials and Methods. We detected specific bands from the FLAG–FANCD2 lysate that were not in the control. In total, we identified 135 proteins as candidate FANCD2-interacting proteins (Tables S1, S2). Previously, numerous proteins have been isolated as FANCD2-binding factors, many of which were screened under DNA damage conditions using DNA-damaging agents (Liu et al., 2010; Smogorzewska et al., 2007; Unno et al., 2014). However, the candidate proteins identified in our screening did not contain previously reported FANCD2-interacting proteins, including FANCI. This finding can be attributed to the lack of exposure to exogenous genotoxic stress in our screening. Consistent with this finding, FANCD2 and FANCI have been reported to form a weak complex under standard immunoprecipitation conditions (Sareen et al., 2012).

To explore the potential biological processes associated with the candidate proteins, gene ontology (GO) enrichment analysis was performed using DAVID Bioinformatics Resources (Sherman et al., 2022). Remarkably, the top-ranked annotation cluster was associated with fatty acid and lipid metabolism or biosynthesis (Fig. 1A). There is increasing evidence for a link between FANC deficiency and changes in lipid homeostasis (Degan et al., 2019). Notably, more than 50% of individuals with Fanconi anemia are diagnosed with dyslipidemia associated with glucose intolerance or hyperglycemia (Petryk et al., 2015). To elucidate the direct role of FANCD2 in cellular lipid metabolism, a *FANCD2* knockout (FANCD2$^{KO}$) cell line was generated from U2OS cells (Fig. S1C,D), and the total fatty acid content in the cells was measured. As shown in Fig. 1B, the total fatty acid content in FANCD2$^{KO}$ cells was significantly lower than that in the parental U2OS cell line and was partially restored by the ectopic expression of wild-type FANCD2. The levels of stearic acid (18:0) among saturated fatty acids and 11-eicosenoic acid (20:1n-9) erucic acid (22:1n-9) and arachidonic acid (20:4n-6) among unsaturated fatty acids showed a significant decrease in FANCD2$^{KO}$ cells and restoration in the complemented cells, respectively. The other fatty acids tested in the present study showed similar patterns, but the differences were not statistically significant.

### Dynamics of FANCD2 in response to OA exposure

The molecular mechanism underlying the association between FANCD2 and fatty acid metabolism remains poorly understood. To

test the hypothesis that FANCD2 functions in not only the DNA damage response but also fatty acid metabolism, we analyzed the subcellular localization of EGFP-FANCD2 under conditions of increased fatty acid levels (Fig. 2A,B). In a culture medium supplemented with OA, most of the FANCD2 retained its localization in the nucleus, as observed under normal culture conditions. However, live-cell imaging captured the atypical nuclear dynamics of FANCD2 after OA treatment. Confocal microscopy showed that a portion of FANCD2 exhibited 'ring-shaped' localization. In some cells, the ring-shaped FANCD2 grew gradually from small dots (Fig. 2A). In another cell, the ring-shaped FANCD2 gradually disappeared (Fig. 2B). The dynamics of the ring-shaped FANCD2 were reminiscent of nuclear lipid droplet (nLD) formation upon OA treatment. Under OA culture conditions, nLD formation occurs rapidly in U2OS cells (Fig. S1E) (Lee et al., 2020; Ohsaki et al., 2016; Sołtysik et al., 2021). Double fluorescence labeling revealed unique subcellular localization of FANCD2 around nLDs in the OA-supplemented medium (Fig. 2C). This LD localization of FANCD2 was enhanced in a dose-dependent manner (Fig. 2D), and its maximal induction was detected after 2 days of OA treatment (Fig. 2E). Moreover, some LDs with FANCD2 were detected even in the cytoplasm (Fig. S2A). It is difficult to distinguish whether these LDs with FANCD2 were observed as a result of *de novo* localization of FANCD2 in cytoplasmic LDs or sustained localization of FANCD2 in nLDs beyond cell division. It has previously been reported that nLDs are extruded into the cytoplasm after mitosis (Sołtysik et al., 2019); thus, FANCD2 might also maintain its localization in nLDs. To examine the nLD localization of endogenous FANCD2, immunofluorescence analysis was performed using anti-FANCD2 antibody. The results of this analysis also demonstrated the nLD localization of endogenous FANCD2 in the OA-supplemented medium (Fig. 2F). Furthermore, correlative light–electron microscopy (CLEM) showed that LDs containing FANCD2 obviously existed in the nucleoplasm and were not cytoplasmic LDs inside the invagination of the nuclear membrane (Fig. 2G,H). No membranous structures were observed on the surface of LDs with FANCD2. Notably, some LDs with FANCD2 interacted with nucleoli or condensed chromatin structures (Fig. 2Hiv; Fig. S2Ciii), whereas others did not (Fig. 2Hiii; Fig. S2Cii). We and others have previously found that nLDs are often associated with promyelocytic leukemia (PML) proteins and PML body components (McPhee et al., 2022; Lee et al., 2020; Ohsaki et al., 2016). PML bodies are subnuclear domains that interact with chromatin and regulate various cellular responses, such as gene expression (Kurihara et al., 2020). Moreover, nLDs are reported to be involved in the transcriptional regulation in yeast cells and mammalian cells (Romanauska and Köhler, 2018; Umaru et al., 2023). These findings prompted us to verify the nuclear localization of FANCD2 and PML following OA treatment. Double-labeled immunofluorescence staining showed that FANCD2 and PML coexisted on some nLDs after OA treatment (Fig. S2D). However, further analysis is required to determine whether FANCD2 localization affects the association of nLDs with chromatin and subsequently modify gene expression.

### OA-induced nuclear lipid droplet formation does not stimulate the DNA damage response

To examine whether the induction of nLD formation with OA leads to a DNA damage response, the phosphorylation of histone H2AX at serine 139 (γH2AX), a molecular marker of DNA damage (Mah et al., 2010), was detected by immunoblot analysis. As shown in Fig. 3A, OA treatment had no effect on the level of γH2AX, unlike

**A**

| Annotation Cluster 1 (Enrichment Score 1.41) | |
| --- | --- |
| Fatty acid metabolism | HACD3, ALDH3A2, FADS2, SLC27A4, SCD5, SCD |
| Fatty acid biosynthesis | HACD3, FADS2, SCD5, SCD |
| Lipid biosynthesis | HACD3, FADS2, SCD5, SCD |
| Lipid metabolism | HACD3, ALDH3A2, FADS2, SPTLC1, SLC27A4, SCD5, SCD |
| **Annotation Cluster 2 (Enrichment Score 0.67)** | |
| Innate immunity | AXL, HK1, HLA-A, HLA-B, METTL3, PRKDC, SIGLEC10, TMEM33 |
| Immunity | AXL, HK1, HLA-A, HLA-B, METTL3, PRKDC, SIGLEC10, TMEM33 |
| Adaptive immunity | HLA-A, HLA-B, SIGLEC10 |
| **Annotation Cluster 3 (Enrichment Score 0.54)** | |
| DNA damages | FANCD2, DDB1, METTL3, PRKDC, RIF1, SMC3 |
| DNA repair | FANCD2, DDB1, PRKDC, RIF1, SMC3 |
| Cell cycle | FANCD2, CLTC, CLTA, PARD3, RIF1, SMC3 |
| **Annotation Cluster 4 (Enrichment Score 0.24)** | |
| Cell cycle | FANCD2, CLTC, CLTA, PARD3, RIF1, SMC3 |
| Cell division | CLTC, CLTA, PARD3, SMC3 |
| Mitosis | CLTC, CLTA, SMC3 |

**B**

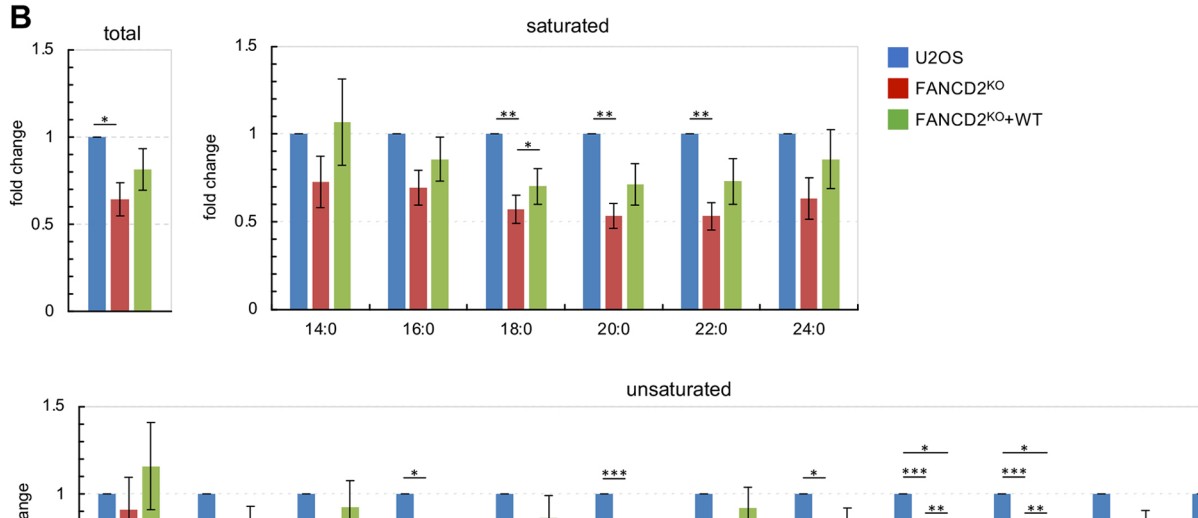

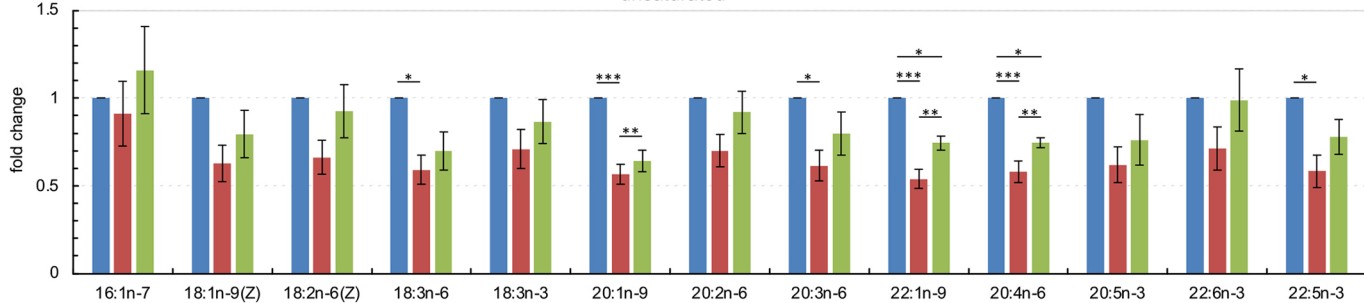

**Fig. 1. Evaluation of cellular fatty acid levels in FANCD2$^{KO}$ cells.** (A) Functional annotation clustering by biological process. The enrichment score represents the geometric mean (in $-\log_{10}$ scale) of the $P$-value in the corresponding annotation cluster. A score >1.3 is considered to be a significant cluster ($P$<0.05). (B) Cellular fatty acid levels were analyzed using GC-MS. Fatty acids detected in this analysis include those derived from phospholipids, triacylglycerols and cholesterol esters, as well as free fatty acids. The relative levels of each fatty acid (saturated and unsaturated) and total fatty acids are shown (mean±s.e.m. from four independent experiments). *$P$<0.05, **$P$<0.01, ***$P$<0.001 (one-way ANOVA coupled with Tukey's multiple comparisons test). Blue, U2OS cells; red, FANCD2$^{KO}$ cells; green, complemented FANCD2$^{KO}$ cells (FANCD2$^{KO}$+WT). Table S3 presents the absolute data.

treatment with the DNA crosslinker mitomycin C (MMC). Consistent with this finding, FANCD2 monoubiquitylation was not induced under the indicated culture conditions (Fig. 3B). In line with previous reports (Liu et al., 2019; Urahama et al., 2008), cell viability under OA treatment (100–400 µM) was not significantly different from that under normal culture conditions (data not shown). Furthermore, immunofluorescence analysis using an anti-

γH2AX antibody showed that γH2AX signals did not colocalize with nLD and FANCD2 in the OA-supplemented medium (Fig. 3C, middle panels). In contrast, MMC treatment had no effect on LD formation, but most nuclear foci of γH2AX colocalized with FANCD2 (Fig. 3C, bottom panels), indicating the sites of MMC-induced DNA damage. The FANCD2 ubiquitin-dead K561R mutant (Fig. S1F), which can no longer be monoubiquitylated by

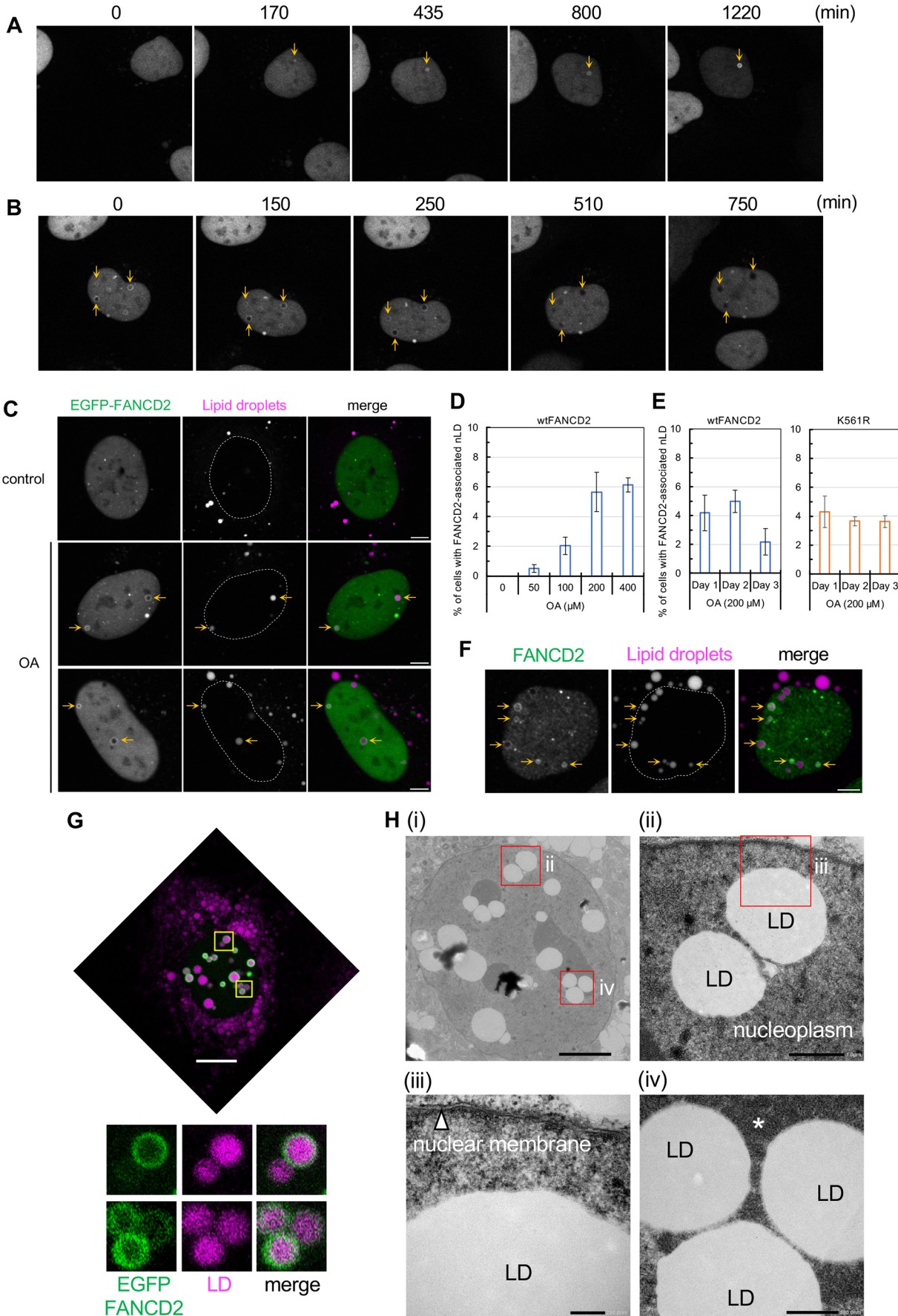

**Fig. 2.** See next page for legend.

**Fig. 2. FANCD2 localizes around nLDs.** (A,B) Images of EGFP–FANCD2 in medium supplemented with 200 µM OA are shown. (A) Growth of EGFP–FANCD2 from small dots to a ring-like appearance (orange arrows). Selected frames from Movie 1 are shown. (B) Disappearance of ring-like EGFP–FANCD2 (orange arrows). Selected frames from Movie 2 are shown. (C) Localization of EGFP–FANCD2 in cells cultured with 200 µM of OA for 2 days. LDs were stained with Lipi-Blue. Orange arrows indicate an nLD with FANCD2. Scale bars: 5 µm. A dashed line indicates the location of FANCD2 (same as nucleus). (D,E) Bar graphs showing the percentage of nLDs with FANCD2 in the cell (mean±s.e.m. from more than three independent experiments). (F) Localization of endogenous FANCD2 in U2OS cells cultured with 200 µM of OA for 2 days. Orange arrows indicate an nLD with FANCD2. Scale bar: 5 µm. A dashed line indicates the location of FANCD2 (same as nucleus). (G,H) CLEM. U2OS cells stably expressing EGFP–FANCD2 were treated with 100 µM of OA for 24 h. (G) Cells were weakly fixed, and fluorescence images were captured first. Some of the nLDs (magenta) were surrounded by FANCD2 (green). Highly magnified images (yellow squares) are shown below. The upper and lower panels correspond to the images in H (ii) and (iv), respectively. Scale bar: 10 µm. (H) (i) Electron microscopy images of the cell shown in G. (ii–iv) Highly magnified images (of the regions indicated by red squares) are shown. Scale bars: 5 µm (i); 1 µm (ii); 200 nm (iii); 500 nm (iv). Asterisk indicates the nucleolar area. Images in F and G,H representative of three repeats.

the Fanconi anemia core complex (Garcia-Higuera et al., 2001), still exhibited nLD localization in the OA-supplemented medium, similar to the wild-type FANCD2 (Fig. 2E). These findings indicate that the nLD localization of FANCD2 is independent of the canonical DNA damage response of FANCD2.

### OA-induced nuclear lipid droplet localization of FANCI

In response to DNA damage, monoubiquitylated FANCD2 and FANCI form a stable heterodimeric complex (ID complex) that targets the site of DNA damage (Ishiai et al., 2017). To examine whether FANCI and FANCD2 colocalize around nLDs after OA treatment, we generated cells stably expressing each protein labeled with different fluorescent proteins, mCherry and EGFP, respectively (Fig. S1G). Consistent with previous findings (Smogorzewska et al., 2007), in response to MMC exposure, FANCD2 formed nuclear foci, which were consistent with those of FANCI (Fig. 4A, bottom panels). By contrast, unlike FANCD2, FANCI exhibited no nLD localization under the OA condition (200 µM, 2 days) (Fig. 4A, middle panels). However, nLD localization of FANCI was detected after FANCD2 depletion in the same cells (Fig. 4B–D). These results indicate that both FANCD2 and FANCI can localize to nLDs, but FANCD2 has a higher tendency to localize to nLDs. The findings suggest that, unlike during the DNA damage response, the two proteins do not stably interact on nLDs to form an ID complex. Consistent with this notion, OA treatment did not promote the complex formation of FANCD2 with FANCI (Fig. S3A), unlike MMC treatment (Fig. S3B). Instead, it is likely that their localization to nLDs is competitive (see below).

### FANCD2 is localized to nLDs in a FANCC-dependent manner

In the DNA damage response, FANCD2 requires the Fanconi anemia core complex to regulate downstream DNA repair process (Ishiai et al., 2017). To explore whether the localization of FANCD2 to nLDs requires Fanconi anemia core complex, FANCC, a component of the Fanconi anemia core complex, was depleted in cells stably expressing EGFP–FANCD2 (Fig. 4E). FANCC depletion diminished the nuclear foci formation of FANCD2 upon MMC exposure, indicating a defect in the DNA damage response of the Fanconi anemia core complex (Fig. S3C). Interestingly, FANCC-depleted cells showed significant reduction

of nLD localization of FANCD2 after OA exposure (Fig. 4F). FANCD2 ubiquitylation is not essential for nLD localization (Fig. 2E). Therefore, these data suggest that FANCC might promote the recruitment of FANCD2 to nLDs, independently of the DNA damage response. The mechanism by which FANC proteins regulate lipid metabolism remains unclear. However, our results suggest that FANC proteins could be involved in lipid metabolic responses that are distinct from the canonical DNA damage response (Fig. S3D).

### Mobility of FANCD2 on nuclear lipid droplets

Two types of subnuclear localization of FANCD2 have been identified – DNA-damage-induced nuclear foci formation and OA-induced nLD localization. However, how stable both of these types of localization are remains unclear. To investigate the mobility of the nuclear-localized FANCD2, fluorescence recovery after photobleaching (FRAP) analysis was performed using cells stably expressing EGFP–FANCD2 (Fig. 5). Nucleoplasmic FANCD2 (no subnuclear localization) showed rapid recovery kinetics under both culture conditions, whereas FANCD2 on nLDs and DNA-damage-induced FANCD2 foci showed significantly lower recovery kinetics. Collectively, these findings demonstrate that FANCD2 undergoes dynamic changes in subnuclear localization in response to OA treatment. Nevertheless, the mobility of FANCD2 around nLDs is limited, indicating a tendency toward stable localization.

### DISCUSSION

Over the past 30 years, the various molecular functions of FANC proteins have been identified. FANC proteins play an important role in the maintenance of genomic stability, particularly in DNA interstrand crosslink damage response and repair. However, although more than half of individuals with Fanconi anemia exhibit impaired endocrine regulation and lipid metabolism, the mechanism by which FANC protein deficiency causes dysregulation remains unclear. Although several studies have shown that the FANCD2 protein interacts with various factors in response to DNA damage, the present findings indicate that FANCD2 interacts with lipid-metabolism-related factors under conditions without exogenous DNA damage (Fig. 1A). FANCD2$^{KO}$ cells showed downregulation of cellular fatty acid levels in normal culture conditions (Fig. 1B). There are two possibilities for the downregulation of cellular fatty acids in FANCD2$^{KO}$ cells; the reduction in fatty acid uptake or the increase in fatty acid consumption. Previously, lipidomic profiling of FANC-deficient cells indicated altered lipid metabolism, particularly a reduction in neutral lipid synthesis and the accumulation of sphingolipids, such as gangliosides and sphingomyelins (Moore et al., 2019; Zhao et al., 2018). In the present study, metabolic profiling was used to detect total cellular fatty acids derived from free fatty acids and fatty acyl chains of neutral lipids that form ester bonds. However, amide-bound fatty acids in sphingolipids were not targeted due to technical limitations (see Materials and Methods). In FANC-deficient cells, intracellular fatty acids might be used for sphingolipid synthesis, resulting in reduced total fatty acid content in neutral lipids. Notably, our GO analysis revealed that the top-ranked annotation cluster was related to lipid and fatty acid metabolism (Fig. 1A). HACD3, FADS2, SCD5 and SCD are involved in fatty acid biosynthesis, whereas SPTLC1 and ALDH3A2 function in the *de novo* synthesis and degradation pathways of sphingolipid metabolism. These results suggest that FANCD2 is involved in the regulation of fatty acid biosynthesis and/or sphingolipid metabolism.

Moreover, a portion of FANCD2 localized around nLDs in the OA-supplemented culture medium (Fig. 2). The ubiquitylation of

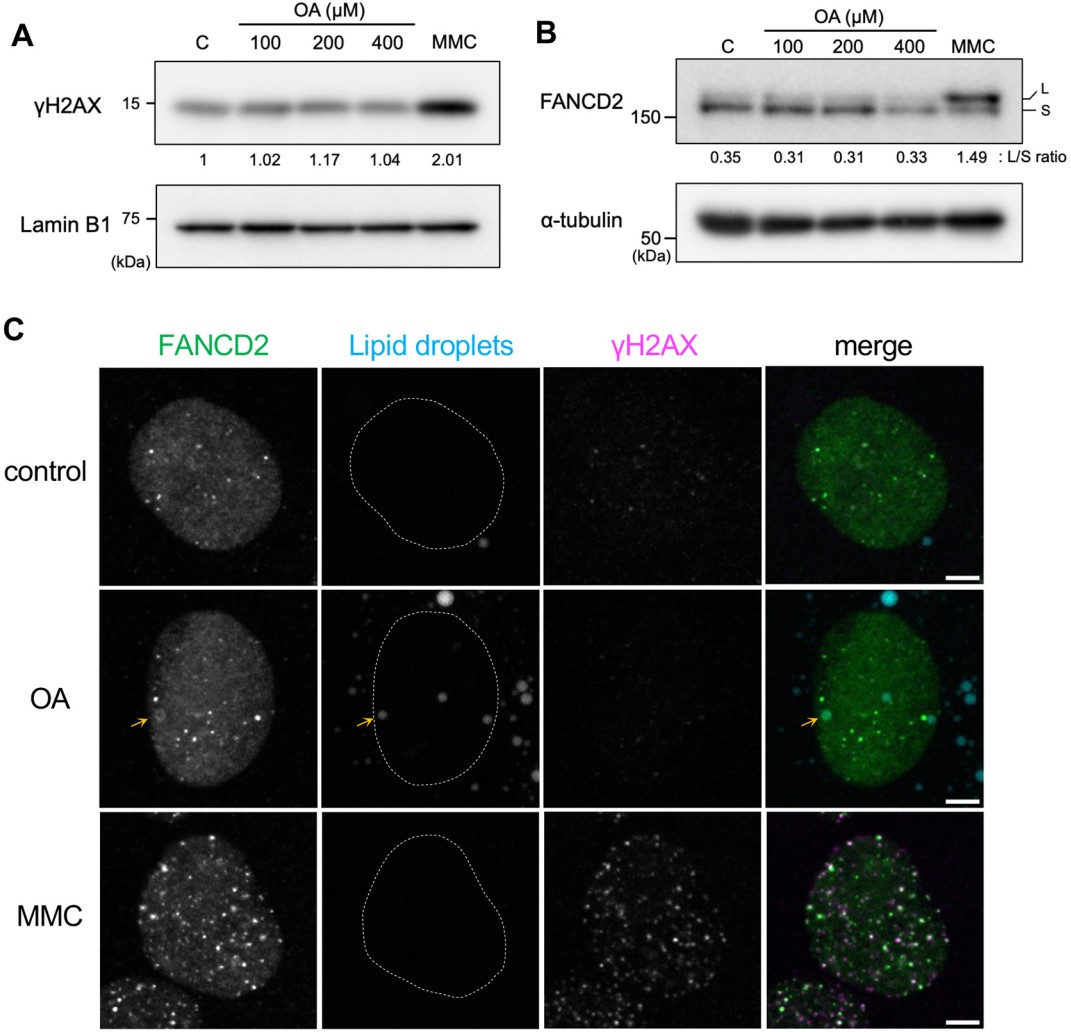

**Fig. 3. nLD localization of FANCD2 is independent of the DNA damage response.** (A) U2OS cells were cultured with the indicated concentration of OA for 2 days. DNA damage was induced by treating cells with 150 nM MMC for 24 h. The fold change in γH2AX levels relative to the control is shown. Lamin B1 was used as a loading control. (B) The conditions for OA and MMC treatment were the same as those described in A. Values below the FANCD2 panel depict the L/S ratios between the monoubiquitylated form (L) and unmodified form (S) of FANCD2. α-tubulin was used as a loading control. Representative blots from more than three repeats are shown. (C) Immunofluorescence analysis of FANCD2 (green) and γH2AX (magenta). LDs (cyan) were stained with Lipi-Blue. Cells were exposed to MMC (150 nM) overnight or OA (200 μM) for 2 days. Orange arrows indicate an nLD with FANCD2. A dashed line indicates the location of FANCD2 (same as nucleus). Representative images from more than three repeats are shown. Scale bars: 5 μm.

FANCD2 was not induced under these conditions (Fig. 3). Consistent with this finding, the ubiquitin-dead K561R mutant exhibited nLD localization (Fig. 2E), suggesting that FANCD2 localization on nLDs is independent of the canonical Fanconi anemia DNA damage response and repair pathway. In line with this notion, the colocalization of FANCD2 and FANCI was not essential for nLD localization (Fig. 4A–D). Furthermore, FANCC depletion suppressed the localization of FANCD2 to nLD (Fig. 4F). These are novel cellular responses of FANCD2 upon exposure to OA independent of the DNA damage response. The biological and physiological significance of FANCD2 in nLDs require further exploration. The canonical cellular location for lipid metabolism is cytoplasmic organelles, such as the endoplasmic reticulum, mitochondria and cytoplasmic LDs. These organelles communicate via local membrane contacts to promote lipid synthesis and lipolysis. Intriguingly, recent studies have reported that lipid metabolism occurs in the inner nuclear membrane (INM) and nLDs of both mammalian and yeast cells (Lee et al., 2020; Romanauska and Köhler, 2018; Sołtysik et al., 2021). In yeast cells, nLD recruits Opi1 – a

transcription suppressor – and regulates lipid metabolism, including phosphatidylinositol lipids (Romanauska and Köhler, 2018). In mammalian cells, some nLDs are enriched in CTP:phosphocholine cytidylyltransferase α (CCTα, a rate-limiting enzyme of *de novo* phosphatidylcholine synthesis pathway) and LPIN1 (a phosphatidic acid phosphatase catalyzing the conversion from phosphatidic acid into diacylglycerol) (Lee et al., 2020; Sołtysik et al., 2019, 2021). Moreover, enzymes involved in lipid metabolism are distributed in the INM and nLDs (Prasad et al., 2011; Vieyres et al., 2020). The present study revealed the stable and dynamic properties of FANCD2 in association with nLD formation, suggesting a novel possibility that FANCD2 is related to fatty acid and/or lipid metabolism in the nucleus (Fig. S3D).

Nevertheless, this study has three limitations that can be addressed in future studies. First, this study focused on the cellular dynamics of FANCD2 following OA exposure. However, it remains unknown whether other FANC proteins localize around nLDs, similar to FANCD2. Second, although FANCD2 localizes around nLDs upon OA exposure, all nLDs in the same nucleus do

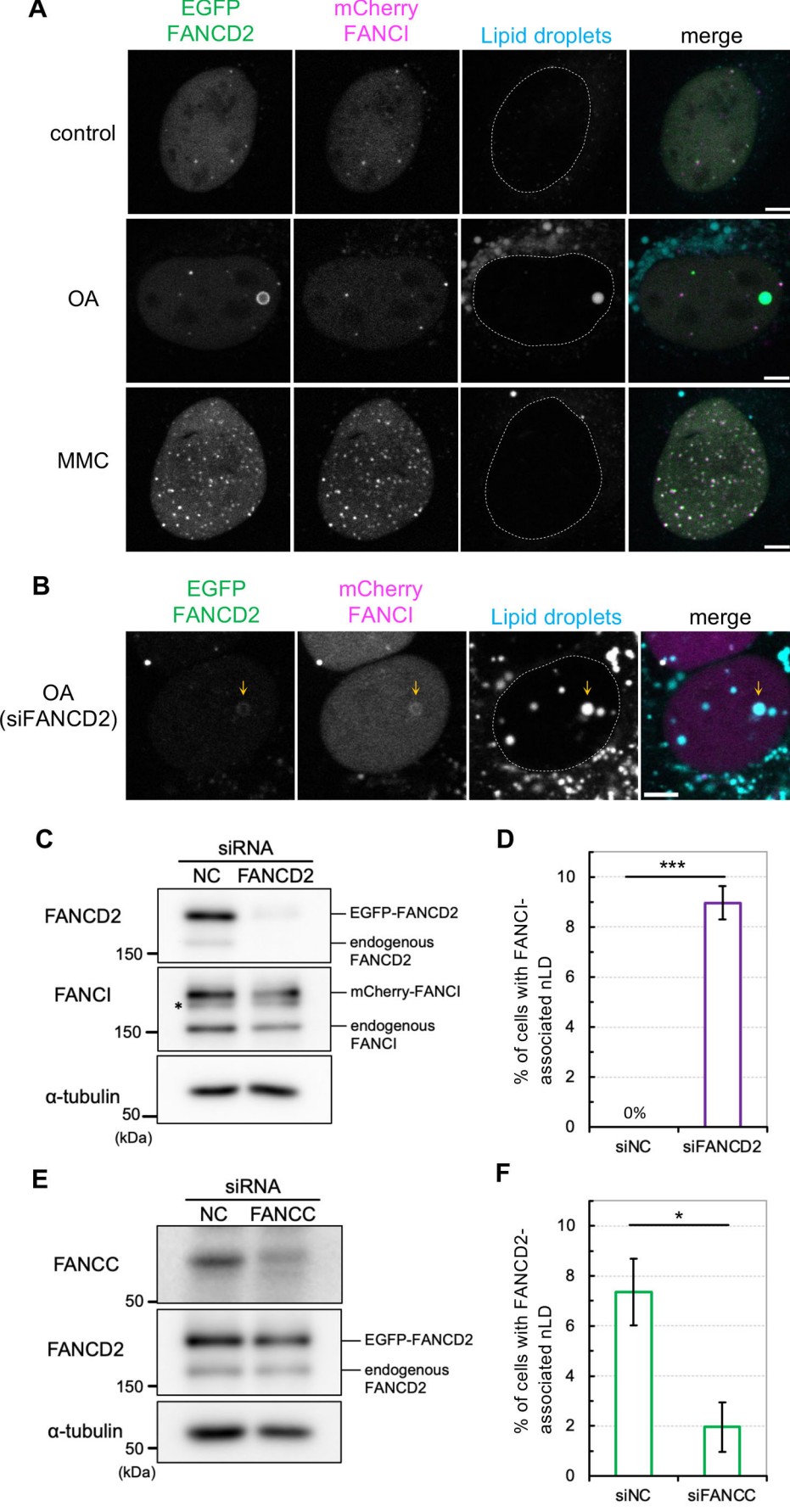

**Fig. 4. OA-induced nLD localization of FANCI.** (A) Live-cell imaging of U2OS cells expressing both EGFP–FANCD2 and mCherry–FANCI. LDs were stained with Lipi-Blue. Representative images from more than three repeats are shown. Scale bars: 5 μm. The conditions for OA and MMC treatment were the same as those described in Fig. 3C. A dashed line indicates the location of FANCD2 or FANCI (same as nucleus). (B) LD localization of FANCI after FANCD2 depletion. The culture conditions were the same as those described in Fig. 2C. Orange arrows indicate FANCD2 and FANCI colocalization in the nLD. A dashed line indicates the location of FANCI (same as nucleus). Scale bar: 5 μm. (C) Immunoblot analyses confirming the depletion of FANCD2 proteins. Asterisk indicates monoubiquitylated endogenous FANCI. (D) Bar graph showing the percentage of nLDs with FANCI (mean±s.e.m. from more than three independent experiments). siNC, negative control siRNA. (E) Immunoblot analyses confirming the depletion of FANCC proteins. (F) Bar graph showing the percentage of nLDs with FANCD2 (mean±s.e.m. from three independent experiments). siNC, negative control siRNA. *P<0.05, ***P<0.001 (two-tailed Student's t-test). Representative blots from more than three repeats are shown in C and E.

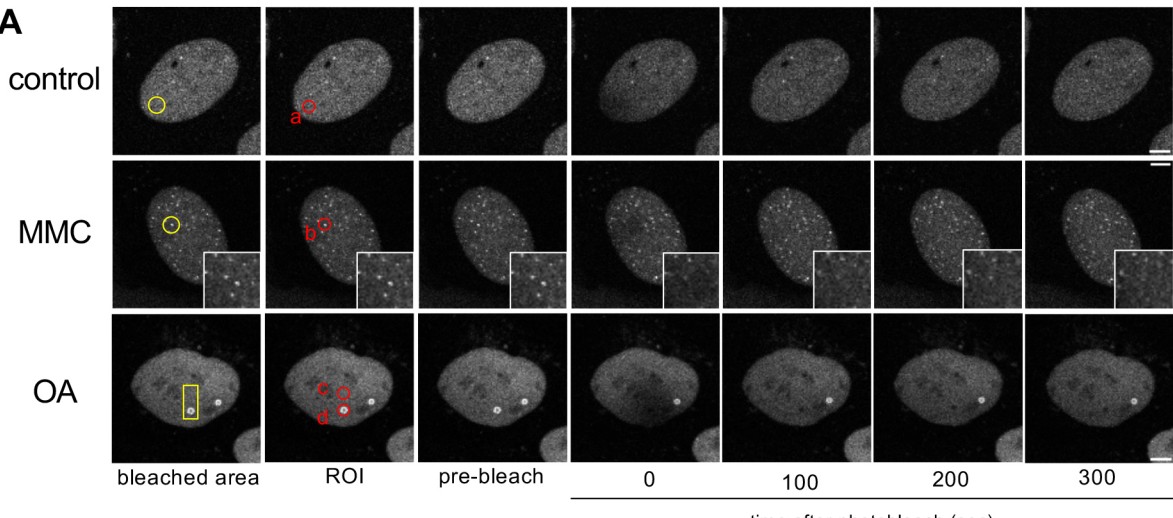

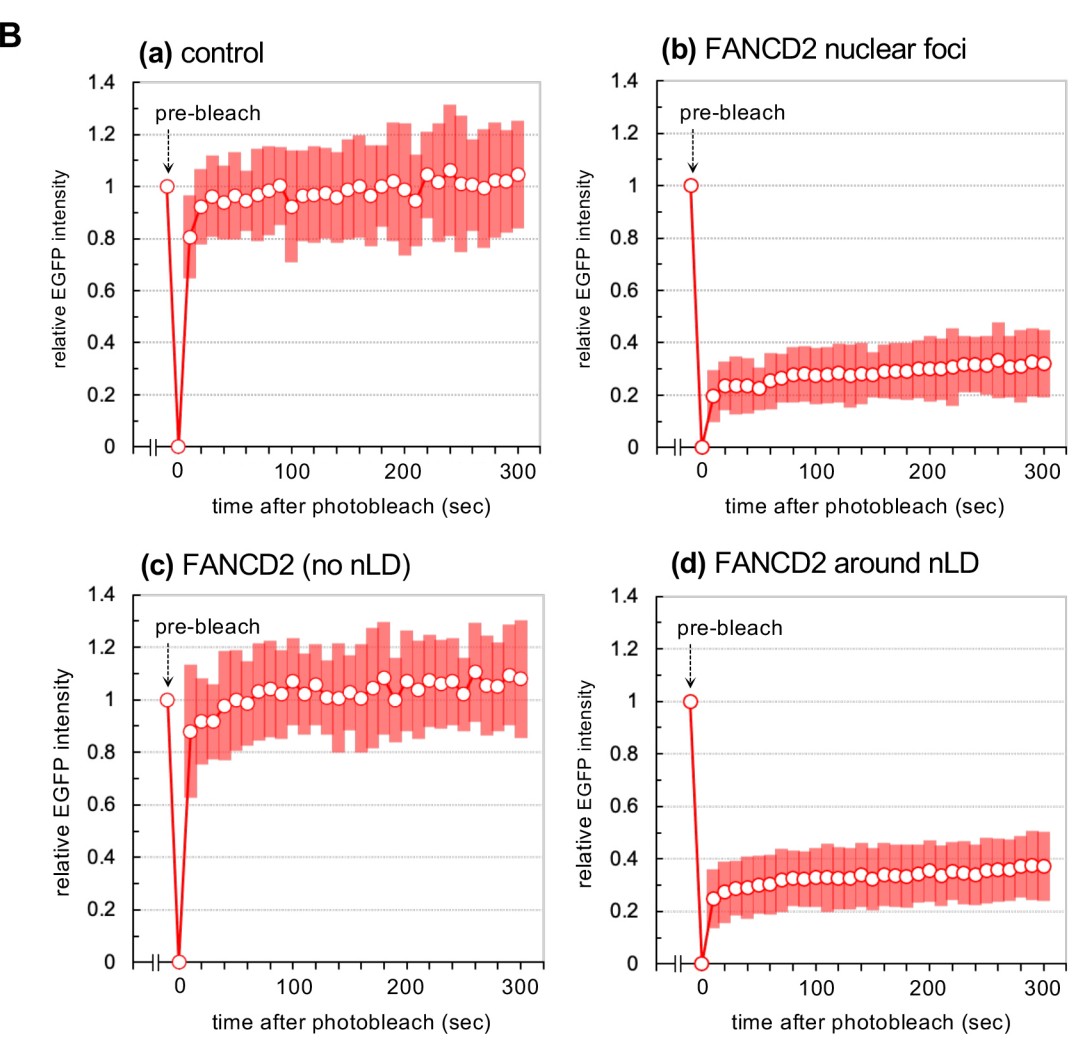

**Fig. 5.** See next page for legend.

not show the presence of FANCD2. Moreover, only a limited portion of cells show FANCD2-associated nLD the same culture conditions. This discrepancy might be explained as follows. First, the interaction of FANCD2 with the nLD surface might be accelerated via unknown partner proteins such as FANCC or suppressed by a competitor such as FANCI. Previously, we and other researchers have shown that the recruitment of CCTα to nLDs is competitively regulated by PLIN3 (Sołtysik et al., 2019) or PML (Lee et al., 2020). Second, the lipid composition on the surface of nLDs might affect the localization of FANCD2. For instance, the

**Fig. 5. Mobility of FANCD2 on nuclear lipid droplets.**
(A) EGFP–FANCD2-expressing U2OS was cultured in the normal medium (control), or medium with 150 nM (MMC) or 200 µM OA, respectively. After the overnight culture, cells were photobleached at the yellow circle or rectangle area (first column). Subsequently, images were captured at the indicated time points. Red circles in the second column represent region of interest (ROI). The highly magnified images in the middle panels (MMC) represent the bleached areas of the FANCD2 foci. The conditions for OA and MMC treatment were the same as those described in Fig. 3C. Scale bars: 5 µm. Selected frames from Movies 3–5 are shown. (B) The fluorescence recovery is shown as relative EGFP intensity based on the initial value before bleaching (pre-bleach) (mean±s.d., $n{\geq}20$; see Fig. S2E,F). (a) Fluorescence recovery after photobleaching (FRAP) analysis of nucleoplasmic FANCD2 under the normal culture conditions (control). (b) FRAP analysis of FANCD2 nuclear focus (+MMC). (c) FRAP analysis of nucleoplasmic FANCD2 (+OA). (d) FRAP analysis of FANCD2 around nLDs (+OA).

amount of phosphatidic acids on the surface of nLDs appeared to vary in minutes (Sołtysik et al., 2021), suggesting that FANCD2 is recruited to nLDs through fluctuating concentrations of lipid-binding partners. Third, another likely explanation is that the size of nLDs defines the association of FANCD2 with nLDs. Smaller nLDs tend to have 'packing defects' in their phospholipid monolayer because of their higher curvature, allowing the proteins to interact with the hydrophobic region of nLDs, and then the proteins may be released as the nLDs grow in size. Finally, recent reports propose that differences in core lipids contents induces phase transition of the LD core between liquid and liquid crystal and will affect the molecular components surrounding LDs (Dumesnil et al., 2023; Mahamid et al., 2019; Shimobayashi and Ohsaki, 2019). These studies suggest that the conditions for phospholipids and/or proteins at the LD surface can vary depending on the rational changes in neutral lipids. Previous lipidomic analysis using rat liver has revealed that nLDs have a higher ratio of cholesteryl esters to triacylglycerols compared to cLDs (Layerenza et al., 2013). Cholesteryl ester-rich LDs can be nucleated at the endoplasmic reticulum by seipin, a key molecule that induces triacylglycerol–seipin clustering, or at the nuclear envelope by SOAT1, a cholesterol esterifying enzyme (Szkalisity et al., 2025). Future lipidomic and proteomic studies will clarify the mechanisms to regulate spatio-temporal differences of LD components. It remains unknown whether the presence of FANCD2 on nLDs affects the function of LDs in lipid metabolism. FANCD2 on nLDs might directly regulate the activity of lipid metabolism enzymes identified as candidate FANCD2-binding partners in this study. Alternatively, these proteins might regulate the functions of transcriptional factors around nLDs, as previously reported (Romanauska and Köhler, 2018; Umaru et al., 2023). Other groups have reported that the biogenesis and cellular localization of cytoplasmic LD are associated with cell cycle progression (Cruz et al., 2019). Elucidating the turnover of FANCD2 on nLDs in the cell cycle might provide a clue to understanding the physiological function of FANCD2 on nLDs. To overcome these limitations, the molecular mechanisms of FANC proteins in lipid metabolism must be investigated in detail. Further research on FANCD2 and other FANC proteins in nLDs will elucidate one of the fundamental questions in the pathophysiology of Fanconi anemia.

## MATERIALS AND METHODS
### Cell lines and cell culture
The human osteosarcoma cell line U2OS (ATCC, HTB-96) and its derivatives were cultured at 37°C in a humidified atmosphere containing 5% $CO_2$ in Dulbecco's modified Eagle's medium (Shimadzu Diagnostics) supplemented with 10% fetal bovine serum (Corning). U2OS cells are suitable for the analysis of DNA damage response and DNA repair because they express wild-type p53. Moreover, they have been widely used to study nuclear lipid droplets and lipid metabolism. The retroviral expression pMMP vector was used for the stable expression of FLAG–FANCD2, EGFP–FANCD2 and mCherry–FANCI (Sakai and Sugasawa, 2014). The pMMP empty vector was used to establish the control cell line (+vec). The production of recombinant retroviruses and infection were performed as previously described (Sakai and Sugasawa, 2014). A culture medium containing 1 µg/ml puromycin (Merck) and/or 10 µg/ml hygromycin (Fujifilm Wako) was used to select stable transformants. Mitomycin C was purchased from Fujifilm Wako (139-18711). Mitomycin C sensitivity of cells was determined by Crystal Violet assay as previously described (Sakai et al., 2008).

### Preparation of the FANCD2 complex
All procedures were performed at 4°C or on ice. To prepare the FANCD2 complex, cell extracts were prepared with CSK buffer (10 mM PIPES-NaOH pH 6.8, 3 mM $MgCl_2$, 1 mM EGTA, 150 mM NaCl, 10% glycerol, 0.1% Triton X-100 and 50 mM NaF) containing protease inhibitors (0.25 mM phenylmethylsulfonyl fluoride, 1 µg/ml leupeptin, 2 µg/ml aprotinin, 1 µg/ml pepstatin, 50 µg/ml AEBSF; Fujifilm Wako). After incubation for 60 min, the cell lysates were centrifuged for 10 min at 20,000 *g* to obtain soluble extracts. The FANCD2 complex was immunoprecipitated from the soluble extracts by incubating with anti-DYKDDDDK tag antibody beads (Fujifilm Wako) overnight with rotation. After extensive washes with CSK buffer, the bound proteins were eluted from the beads via incubation for 60 min with 0.5 mg/ml FLAG peptide (Fujifilm Wako) in the same buffer. After ammonium sulfate precipitation, the precipitated proteins were dissolved in SDS sample buffer (62.5 mM Tris-HCl pH 6.8, 10% glycerol, 1% SDS and 1% 2-mercaptoethanol) and dialyzed with Biotech CE MWCO 8000 (Bio-tech) in the same buffer. Samples were separated by SDS-PAGE, and the CBB-stained bands were excised and digested in-gel with trypsin.

### Nano-liquid chromatography–mass spectrometry and database search
The desalted samples were then subjected to nano-liquid chromatography–mass spectrometry (LC-MS) analysis using a Q-Tof Ultima API (Waters) coupled to a NanoFrontier nLC (Hitachi). The peptides were resolved by reversed-phase chromatography using a capillary column made in-house (10 cm long, 50 µm in diameter) packed with 3 µm silica, using ReproSil-Pur C18-AQ resin (Dr Maisch GmbH). Peptides were eluted as follows (solvent A, 2% acetonitrile and 0.1% formic acid; solvent B, 98% acetonitrile and 0.1% formic acid; flow rates: 200 nl/min): 0–5 min, linear gradient from 2% to 5% B; 5–60 min, linear gradient to 23% B; 60–65 min, linear gradient to 70% B; 65–70 min, isocratic with 70% B; 75–75.1 min, linear gradient to 2% B; 75.1–105 min, isocratic with 2% B. MS was performed in positive mode, and spectra were acquired from 300 to 1800 $m/z$ for 1 s followed by three data-dependent MS/MS scans from 45 to 900 $m/z$ for 2 s each. The collision energy used to perform MS/MS was automatically varied according to the mass and charge state of the eluting peptide. Only the spectra of ions with charge states 2, 3 and 4 were acquired for the MS/MS analysis. The dynamic exclusion duration was set to 60 s. The typical mass spectrometric conditions were as follows: capillary voltage, 3.2 kV; source temperature, 80°C. Matrix Science Mascot Distiller and Mascot Server were used for peptide peak picking and protein identification, respectively. Database search was performed against the human International Protein Index protein database (version 3.53) with the following parameters: enzyme, trypsin; missed cleavages, 2; fixed modification, carbamidomethyl (C), variable modification, methionine oxidation, pyroglutamylation (N-terminal E); peptide length, more than 6. A peptide mass tolerance of 100 ppm and fragment mass tolerance of 0.3 Da were used. The false discovery rate was set to <0.05. The screening of the FANCD2-binding partners in this study is the result from a single preparation of the FANCD2 complex. Gene ontology analysis was performed using DAVID Bioinformatics Resources (Sherman et al., 2022). The enrichment score is equivalent to $-\log_{10}$ *P*-value.

## Gene disruption of *FANCD2*

Disruption of the endogenous *FANCD2* gene in the U2OS cell line was performed using the GeneArt CRISPR Nuclease Vector with CD4 Enrichment Kit (Thermo Fisher Scientific). The gRNAs targeted sequences within exon 11 of FANCD2 (5′-GTTGTCGTCTATTAGATTGG-3′). After enrichment with anti-CD4 magnetic beads, single clones were isolated by limiting dilution. FANCD2 expression in each clone was assessed using immunoblotting, and gene disruption was confirmed using Sanger sequencing of genomic DNA.

## Total fatty acid analysis with gas chromatography-mass spectrometry

For gas chromatography-mass spectrometry (GC-MS) analysis, cells were collected with 70% methanol after two rinses with PBS. The protein concentration in each cell extract was determined and normalized. Nonadecanoic acid (C19:0, 5 nmol) was added as an internal standard to each sample. Total fatty acid methyl esters were prepared using a fatty acid methylation and purification kit (Nacalai Tesque) according to the manufacturer's instructions. In summary, total fatty acids were extracted using hexane. The esterified fatty acids (from phospholipids, triacylglycerol and cholesteryl esters) were saponified and methylated by a sodium hydroxide and methanol solution. Free fatty acids were methylated by a hydrochloric acid and methanol solution. After concentration by evaporation, the methyl ester-derivatized fatty acid was reconstituted with 100 μl of hexane for subsequent analysis. Fatty acids were analyzed using GC-MS (QP2010 Ultra, Shimadzu). The capillary column used for fatty acid separation was SP-2650 (100 m length×0.25 mm inner diameter×0.20 μm film thickness, Sigma-Aldrich). The column oven temperature was increased from 140°C to 240°C, and the separated fatty acid methyl ester was detected using mass spectrometry. The standard mixture of methyl ester fatty acids was obtained from Sigma-Aldrich.

## Lipid droplet induction and staining

OAs (Merck) were conjugated with fatty acid-free bovine serum albumin (Fujifilm Wako) as previously described (Ohsaki et al., 2016). For the staining of LDs, Lipi-Blue or Lipi-Deep Red (Dojindo Laboratories) was used according to the manufacturer's instructions.

## Immunofluorescence

Cells were seeded in 35 mm glass-bottom dishes (Matsunami Glass) and incubated under the conditions indicated in each figure legend. Fixation and immunofluorescence staining were performed as previously described (Sakai and Sugasawa, 2014). Antibodies against the following proteins were used in this study: FANCD2 (1:500; NB100-182, Novus Biologicals), phospho-H2AX (1:500; 05-636, Merck) and PML (1:500; sc-966, Santa Cruz Biotechnology). Alexa Fluor secondary antibodies were purchased from Thermo Fisher Scientific (1:1000; A-11020, A21206). Single plain images were acquired on Olympus FV3000 a confocal laser scanning microscope (Evident) using a UPlanSApo 40× lens with a numerical aperture (NA) of 0.95.

## Immunoblot analysis

Protein extracts were prepared with CSK buffer (10 mM PIPES-NaOH pH 6.8, 3 mM MgCl$_2$, 1 mM EGTA, 300 mM NaCl, 10% glycerol, 0.1% Triton X-100, 50 mM NaF) containing protease inhibitors (0.25 mM phenylmethylsulfonyl fluoride, 1 μg/ml leupeptin, 2 μg/ml aprotinin, 1 μg/ml pepstatin, 50 μg/ml Pefabloc SC). After incubation on ice for 30 min, the cell lysates were centrifuged for 10 min at 20,000 *g* to obtain soluble extracts. The insoluble fractions were resuspended in the same buffer via sonication and used to detect γH2AX and Lamin B1 in immunoblot analysis. The protein concentration was measured using an XL-Bradford assay kit (Apro Science). Immunoblotting was performed as previously described (Sakai et al., 2020). Antibodies against the following proteins were used in the present study: FANCD2 (1:200; sc-20022, Santa Cruz Biotechnology), FANCI (1:400; ab74332, Abcam), FANCC (1:500; TA325105, OriGene Technologies), phospho-H2AX (1:2000; A300-081A, Fortis Life Sciences; 05-636, Merck), α-tubulin (1:20,000; T5168, Merck), Lamin B1 (1:1000; C-20, Santa Cruz Biotechnology) and FLAG (1:1000; PM020, Medical & Biological

Laboratories). Alkaline phosphatase-conjugated secondary antibodies were purchased from Merck (1:20,000; A3562, A3812 and A4062). Uncropped images of western blots from this paper are shown in Fig. S4.

## Treatment with siRNA

The target sequence for the depletion of FANCD2 has been described previously (Liu et al., 2010), and siRNA was synthesized by Japan Bio Services. siRNA targeting FANCC was purchased from Horizon Discovery (siGENOME Human FANCC siRNA, SMARTpool). The control siRNA (AllStars Negative Control siRNA) was purchased from Qiagen. Cells were transfected with siRNA using the Lipofectamine RNAiMAX reagent (Thermo Fisher Scientific), and the final concentration of siRNA in the culture medium was adjusted to 40 nM (FANCD2) or 20 nM (FANCC).

## Fluorescence recovery after photobleaching

U2OS cells expressing EGFP–FANCD2 were seeded in 35 mm glass-bottom dishes (glass diameter, 27 mm; glass thickness No. 1S; poly-lysine coated; Matsunami Glass) and cultured under the conditions indicated in the legend. Under an FV3000 confocal laser scanning microscope (Evident) equipped with a UPlanSApo 40× lens with a 0.95 NA, EGFP fluorescence was bleached with a 488 nm laser at 100% power. After bleaching, fluorescence images were acquired every 10 s for 5 min.

## Correlative light–electron microscopy

U2OS cells stably expressing EGFP–FANCD2 were cultured on gridded glass coverslips (Matsunami Glass) and treated with 100 μM of OA for 24 h. After labeling with Lipid-Tox Red (Thermo Fisher Scientific) to visualize LDs, the cells were fixed with 4% paraformaldehyde and 0.05% glutaraldehyde in 0.1 M phosphate buffer for 30 min. Fluorescence images were captured by an A1 confocal laser scanning microscope (Nikon) equipped with a GaASP multidetector using a PlanApo 100× lens with a 1.45 NA. Cells were fixed with 2.5% glutaraldehyde in 0.1 M sodium cacodylate buffer (pH 7.4) for 2 h, postfixed with 1% osmium tetroxide and 0.1% potassium ferrocyanide for 1 h, dehydrated, and embedded in epoxy resin. Ultrathin sections were cut and observed using a JEM1400 (JEOL) operated at 80 kV.

## Quantification and statistical analysis

Quantification of the immunoblot analysis was performed using GeneQuant (Cytiva). All signals were normalized by the intensity of α-tubulin or Lamin B1. Statistical analyses were performed as indicated in the figure legends using GraphPad Prism 9 (GraphPad Software).

### Acknowledgements
The authors are grateful to the members of Biosignal Research Center and the Department of Biology, Graduate School of Science, Kobe University, for helpful discussions and encouragement, and to the Research Facility Center for Science and Technology of Kobe University for DNA sequence analysis.

### Competing interests
The authors declare no competing or financial interests.

### Author contributions
Conceptualization: Yuki Ohsaki, W.S.; Formal analysis: T.H., M.G., Yukie Otsuki, S.M., N.W., M.S.; Funding acquisition: Yuki Ohsaki, W.S.; Investigation: T.H., M.G., Yukie Otsuki, S.M., N.W., M.S., Yuki Ohsaki; Methodology: T.H., M.G., Yukie Otsuki, S.M., N.W., M.S.; Project administration: W.S.; Resources: T.M., M.Y., K.S.; Supervision: T.M., M.Y., K.S., Yuki Ohsaki, W.S.; Visualization: T.H., Yuki Ohsaki, W.S.; Writing – original draft: T.H., Yuki Ohsaki, W.S.; Writing – review & editing: T.H., Yuki Ohsaki, W.S.

### Funding
This work was supported by a Grant-in-Aid to W.S. (Japan Society for the Promotion of Science KAKENHI grant number 17K07286 and 20K06487), and by a Grant-in-Aid to Yuki Ohsaki (Japan Society for the Promotion of Science KAKENHI Grant Number 21K06733, 24K02208, and Takeda Science Foundation). Open access funding provided by by SPRING (JST). Deposited in PMC for immediate release.

### Data and resource availability
All relevant data and details of resources can be found within the article and its supplementary information. Any cell line, and plasmid DNA used in this study is available from the lead contact upon request.

**Peer review history**
The peer review history is available online at https://journals.biologists.com/jcs/lookup/doi/10.1242/jcs.264430.reviewer-comments.pdf

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
