## [Peer Review File · Journal of Cell Science]

Dynamics of Fanconi anemia protein D2 in association with nuclear lipid droplet formation

Tomoya Hotani, Motonari Goto, Yukie Otsuki, Shun Matsuda, Nobuhiro Wada, Masakazu Shinohara, Tomonari Matsuda, Masayuki Yokoi, Kaoru Sugasawa, Yuki Ohsaki and Wataru Sakai

DOI: 10.1242/jcs.264430

Editor: Megan King

Review timeline

Submission to Review Commons:	4 February 2025
Submission to Journal of Cell Science:	5 September 2025
Accepted:	12 September 2025

Reviewer 1

Evidence, reproducibility and clarity

Summary

Fanconi anemia is characterized by dysfunction of DNA damage repair. The authors found FANCD2, one of gene products responsible for Fanconi anemia, out a study of SDF2L1, is localized on nuclear lipid droplets and is involved in cellular metabolism of fatty acids.

Major comments

1. Why the author designed to use U2OS cells in this study? U2OS cell line is an epithelial cell originated from osteosarcoma. I wonder whether it is a suitable cell line to study lipid metabolism and/or lipid droplet accumulation.
2. Figure 1B. The authors found that cellular fatty acids reduced in FANCD2-KO cells.
 - 2-1. What do you think from this observation is happened in the cells? Is it a reduction of fatty acid uptake by the cells, or it is an increase of fatty acid consumption?
 - 2-2. Free fatty acids are not necessarily the major lipids in cells. Phospholipids, triacylglycerol and cholesterol, in addition to fatty acids, should be examined.
3. Figure 3. The MMC concentration used for WB experiments A and B was 500 microM, while 150 microM was used for microscopic experiment C. Why the conditions are different.
4. Figure 2D. Why do you think only a limited portion (approximately 6%) of nLD is associated with FANCD2, although the amount of FANCD2 is overwhelmed to endogenous FANCD2 (Fig. S1D).
5. Overall, this manuscript demonstrates a novel feature of FANCD2 protein; this is relatively enriched distribution to nuclei, nLD association upon OA treatment, fatty acid decrease when FANCD2 is depleted. These observations suggest some kind of relation between FANCD2 and lipids. However, I am afraid there is no progress in understanding the roles of FANCD2 on lipid metabolism yet. The last paragraph of "Introduction" and the last two sentences on page 9 should be rewritten.

Specific points.

1. Figure 1A. The explanation of the "enrichment score" is insufficient. It is nice to show the equation to draw the score if it is helpful for the readers to figure out what the score represents. Or, p-values equivalent for enrichment score 0.67, 0.54, 0.24 should be indicated.

2. The abbreviation PML should be spelled out at their first appearance.
3. Figure 5. Video data should be provided to show the photo bleach experiments.
4. The legend for Figure 5 is poor. Experimental conditions should be described to help understanding to the readers. The left panel of Figure 5B may be divided into two separate graphs,
5. Summary, line 7. What do you mean for "lipid metabolic-related factors". If you think of a particular enzyme(s) or pathway, please mention it rather than an ambiguous expression.
6. Introduction, line 8-12. Please cite a relevant reference regarding ID complex formation and its importance on FA pathway.

Significance

This manuscript reports novel observations on FANCD2 protein and its unique behavior in nuclei. However, current data are not enough to elucidate roles of FANCD2 in lipid metabolism. For example, the nuclear protein FANCD2 localizes predominantly in nuclei by interacting with chromatin and other nucleic acids, but it can associate with lipid droplets by disrupting the microenvironment when lipid droplets are formed in nuclei. There is still something missing between the localization of FANCD2 to nLD and lipid metabolism. This manuscript seems relatively difficult to read, and I think it will be nice if further explanations on the figure legends and data curations.

Reviewer 2

Evidence, reproducibility and clarity

Fanconi anemia is an inherited genetic disorder caused by mutations in proteins performing DNA repair, leading to bone marrow failure and increased risk for the development of cancer. Notably, many patients display lipid imbalance, but the molecular mechanism of dysregulations is far from understood. Hotani et al. study the alternative role of one of the components of the effector complex in DNA repair pathway, the FANCD2, more specifically, its role in abnormal lipid metabolism.

To study the independent role of FANCD2 in DNA repair, they used U2OS cells, a cellular model unrelated to Fanconi anaemia, and did not use DNA damage conditions. Co-purification of potential protein partners of FANCD2 provided a list of 135 candidates, and ten were annotated to be involved in lipid metabolism (cluster 1). The follow-up study reveals precise and previously undescribed localization of FANCD2 to nuclear lipid droplets (nLDs) in response to treatment with a commonly used agent in lipid metabolism research, the OA (Oleic Acid). Nuclear localization was confirmed using electron microscopy. Observed nLDs translocation was not related to DNA damage response verified by no increase in phosphorylation of histone H2AX or ubiquitination of FANCD2 upon OA treatment. Interestingly, unlike in DNA repair, metabolic changes do not lead to the formation of a heterodimeric complex of FANCD2 with FANCI on LDs, but they rather compete for it. Altogether, the study presents convincing evidence of a new LD protein. However, its physiological role in lipid metabolism is to be discovered.

Major comments: Authors carefully conclude and present satisfactory data to support their claims.

1. OPTIONAL: Proteomic analysis was performed on whole cell lysate. FANCD2 is a nuclear protein. Co-purification of FANCD2 in the nuclear fraction could give even more accurate proteomic results.
2. OPTIONAL: As the authors discussed in the last part of the manuscript, the question of whether the presence of FANCD2 on nLDs affects LDs' function is still to be answered. Experiments which could shed more light on that issue could be either i) proteomic analysis of co-purified FANCD2 partners under OA treatment - nuclear fraction. ii) alternatively, proteomics of isolated nLDs from nuclear fraction.

Minor comments:

1. To obtain fluorescent images, authors in M&M reveal the use of confocal microscopy. However, it is unclear if the images presented in the main figures are maximum intensity projections or single plain images. It is essential to also include in supplementary data one example of one dataset sequence of confocal images covering an entire volume of cells with DAPI and LDs staining. It will allow the differentiation of the nuclear LDs from the cytoplasmic ones.
2. Authors in the discussion paragraph listed three reasons for the observed heterogeneity in the localization of FANCD2 to just a fraction of nLDs. However, I suggest discussing an additional factor, the LD's core content, which could also affect protein occupancy.
3. In the lipid droplet and lipid metabolism field, the "FA" abbreviation is very often used to describe fatty acids. If possible, I propose to change the Fanconi anemia abbreviation.
4. In the following sentence, it is not clear which "this conditions " are: "However, unlike FANCD2, FANCI exhibited no nLD localization under this condition " I propose to spell out the OA conditions.
5. I propose not to introduce the ICL abbreviation. as it is mentioned only two times in the text.
6. Not correct citation of the figure panel in the following sentence: "To test the hypothesis that FANCD2 functions in not only the DNA damage response but also fatty acid metabolism, we analyzed the subcellular localization of EGFP-FANCD2 (Fig S1D) under conditions of increased fatty acid levels."

Significance

The manuscript presents a clear observation of the specific localization of FANCD2 to nLDs in the condition of fatty acids loading. It is an important study from the perspective of Fanconi anemia as well as the fundamental biology of lipid droplets. The role of the FANCD2 protein in Fanconi anemia is well studied in DNA damage response. In contrast, its specific role in lipid metabolism is to be discovered, and the presented study shows clear links which point to interesting directions to follow. It is also important to have a better understanding of LDs' function in health and disease. Cytoplasmic LDs are significantly better studied than nuclear LDs. Also, the majority of the literature on nuclear LDs is in the field of viral infection. Thus, this study gives an important example that should be verified and tested in other cell types and conditions. Another important aspect described in the study is the apparent heterogeneity in the occupancy of FANCD2 on nLDs. Multi-dimensional heterogeneity of LDs is under active investigation in various models and studies, which can help understand the mechanisms driving specific protein occupancy in just subsets of LDs, which are important to be shared.

The main limitation, also discussed by authors, is missing understanding of the biological function of FANCD2 nLDs localization and should be addressed in the future.

Audience: Basic research; Fanconi anemia researchers; lipid droplets researchers.

Reviewer 3

Evidence, reproducibility and clarity

Summary

In this study, Hotani et al. investigate novel roles in lipid metabolism for FANCD2, which is best known for being mutated in the genomic instability syndrome Fanconi Anemia and for its canonical function as part of the cellular response to interstrand DNA crosslinks. Additional published work indicates that FANCD2 might have other roles in metabolism and here the authors provide evidence that it accumulates at a subset of nuclear lipid droplets, further suggesting non-canonical roles in lipid metabolism. In a screen for FANCD2 interacting proteins in U2OS cells without genotoxin treatment, the authors observed that proteins involved in fatty acid metabolism and lipid metabolism were significantly enriched as interactors with FANCD2. They then went on to show that FANCD2 deficient cells have altered fatty acid content. Moreover, FANCD2 stably localized to nuclear lipid droplets (nLDs) in response to oleic acid treatment, and it did so independently of its well characterized binding partner, FANCI, and in the absence of markers of the DNA damage response activation.

Major comments

- The authors begin by highlighting 135 potential candidate FANCD2 interactors, but do not provide a full data set with information about the specific peptides observed, the number of times individual peptides were detected, or probability data. The details of biological and technical replicates analyzed in the experiment are unclear to this reader. The detailed proteomics data also should be deposited in an appropriate public database. Although it could be considered beyond the scope of the current manuscript, there is no follow up with characterization (i.e. co-IP validations or colocalization studies using immunofluorescence) for any of the candidate proteins from the screen.
- The manuscript is limited to the analysis of one cancer cell line, U2OS, and it appears one CRISPR edited FANCD2 KO clone. It would be useful to know whether the localization of FANCD2 at nLDs is conserved in other systems other than U2OS cells, including normal non-malignant cells as well as other cancer cell lines. The methods suggest that multiple KO clones were generated and the paper would be strengthened by data confirming that the results are consistent across multiple clones (although importantly fatty acid phenotypes are largely rescued by reintroducing FANCD2 into the KO cells).
- The localization of FANCD2 to nLDs occurs without FANCI, is not stimulated by MMC, and doesn't require FANCD2 ubiquitination. These are interesting data that distinguish this mechanism from the canonical DDR role of FANCD2. The authors cite work showing that metabolic alterations occur in many FA patients (presumably independent of complementation group) and that lipid metabolism is reported to be altered in FANCA mutant cells. It would be worthwhile to test whether the localization of FANCD2 to nLD2 requires any core complex factors or is entirely independent of them. The authors also provide some data suggesting a potential competition between FANCD2 and FANCI at lipid droplets, but overall it is unclear from the study how the proposed role for FANCD2 fits into the FA pathway overall. As noted by the authors as a study limitation, the work also does not address the downstream mechanisms by which FANCD2 modulates lipid metabolism.

Minor comments

- Given the low percentage of nLD with FANCD2 present (Fig. 2D/E), it would be helpful if the total number lipid droplets observed per nuclei in untreated and OA treated U2OS cells was also reported.
- Figure 2A/B: what is the percentage observed between retained FANCD2 localization versus loss of FANCD2 localization over time? Quantitative data and discussion about the turnover of FANCD2 at these droplets would be beneficial. As suggested in the manuscript, it would be worthwhile to explore the possibility of FANCI localizing at these regions as FANCD2 is lost.
- The prior literature is thoroughly and thoughtfully described.
- The figures are put together nicely and the data are generally of very high quality. The writing is clear and logical.

Significance

Given that patients with FA have disruptions in lipid metabolism, this study begins to elucidate how FA proteins, mainly FANCD2, may be involved in lipid metabolic processes. Although lipid droplet accumulation is a known feature in FA patients, this is the first observation of FANCD2 localizing to nuclear lipid droplets (nLDs). The study starts with a proteomic screen identifying 135 potential FANCD2-interacting proteins under "normal" conditions. More robust data reporting and further investigation into FANCD2-associated proteins involved in lipid biosynthesis and metabolism identified in this screen would strengthen the study's findings. Additionally, the authors hint at a potential function for FANCD2 outside its traditional role in DNA damage response and independent of FANCI. Further analysis of other FA proteins-such as their requirement for FANCD2 localization as well as their own localization to nLDs- would also be beneficial. How broadly applicable the observations are remains unknown given that the work is limited to U2OS cells, and the physiological importance of FANCD2 localization to nLDs will require further study. This paper will be of interest to researchers in both the DNA damage and lipid metabolism fields as a brief, initial observation for a non-canonical function for FANCD2 deserving of further study.

Author response to reviewers' comments

Manuscript number: RC-2025-02900

Corresponding author(s): Yuki OHSAKI and Wataru SAKAI

1. General Statements

First of all, we would like to sincerely thank all reviewers for their very valuable and constructive comments, by which we believe that our manuscript has been greatly improved. Our responses to the reviewers' comments are listed below.

Reviewer #1

Major comments

1. Why the author designed to use U2OS cells in this study? U2OS cell line is an epithelial cell originated from osteosarcoma. I wonder whether it is a suitable cell line to study lipid metabolism and/or lipid droplet accumulation.

As reviewer #1 pointed out, U2OS is a cell line derived from human osteosarcoma. Human hepatoma cell line, such as Huh-7 and HepG2, are used to study lipid metabolism and lipid droplet formation. However, our study focuses not only on lipid metabolism but also on DNA damage response. U2OS cell line has functional p53 and is suitable to analyze DNA damage response and DNA repair. Many papers are published using U2OS in research of DNA damage response. Importantly, this cell line has been widely used to analyze nuclear lipid droplet formation (Ohsaki et al, *J Cell Biol* 212, 29-38 (2016). Wong et al, *Cell Death Discovery* 4, 109 (2018). Soltysik et al, *Nat Commun* 10, 473 (2019). Lee et al, *Life Sci Alliance* 3, 8, (2020). Ivanovska et al, *J Cell Biol* 222, 8 (2023)). From these points, we believe that U2OS is advantageous for our research. To address the reviewer's concern, materials and methods in the revised manuscript was modified (page 12 line 6).

2. Figure 1B. The authors found that cellular fatty acids reduced in FANCD2-KO cells.

2-1. What do you think from this observation is happened in the cells? Is it a reduction of fatty acid uptake by the cells, or it is an increase of fatty acid consumption?

It is very important to address what is responsible for the reduction of cellular fatty acid in the FANCD2^{KO} cell. In this paper, we discuss only the latter possibility: increase of fatty acid consumption (page 9 line 22). However, discussion on both points is now included in the revised manuscript (page 9 line 13), since both possibilities are conceivable as the reviewer pointed out.

2-2. Free fatty acids are not necessarily the major lipids in cells. Phospholipids, triacylglycerol and cholesterol, in addition to fatty acids, should be examined.

We agree with the reviewer #1 that only free fatty acid is not a major cellular lipid. First, the metabolic profiling in this study have detected total fatty acids from phospholipids, triacylglycerol, and cholesteryl esters in addition to free fatty acids. To help readers understand this point, the legend to Figure 1B was modified to clearly state that fatty acids from phospholipids, triacylglycerol, and cholesteryl esters are included in this analysis (page 18 line 8), as well as the materials and methods (page 14 line 11). Second, in addition to fatty acids, it is very important to analyze different types of lipids such as neutral lipids, phospholipids, sterols and sphingolipids to elucidate the effects of FANCD2 deficiency on cellular lipids metabolism. We would like to investigate total changes of lipids metabolism in the future work.

3. Figure 3. The MMC concentration used for WB experiments A and B was 500 microM, while 150 microM was used for microscopic experiment C. Why the conditions are different.

We sincerely thank the reviewer #1 for this thoughtful comment. The analyses with different experimental conditions may confuse the readers. Therefore, we performed the analysis under the same conditions (MMC 150 nM) and modified Figure 3AB.

4. Figure 2D. Why do you think only a limited portion (approximately 6%) of nLD is associated with FANCD2, although the amount of FANCD2 is overwhelmed to endogenous FANCD2 (Fig. S1D).

We thank the reviewer #1 for raising this crucial point. This is related to the minor comment (point 2) from reviewer #2. It has been reported that both cytosolic and nucleoplasmic LDs have heterogeneity of proteins and lipids composing each LD in a single cell. We have already described several hypotheses why protein components can be changed on the surface of LDs in page 10. Also, as another aspect suggested by the reviewer #2, the hypothetical relationship between LD core lipids and surface proteins, is now discussed as the 4th hypothesis in the revised manuscript (page 11 line 8).

5. Overall, this manuscript demonstrates a novel feature of FANCD2 protein; this is relatively enriched distribution to nuclei, nLD association upon OA treatment, fatty acid decrease when FANCD2 is depleted. These observations suggest some kind of relation between FANCD2 and lipids. However, I am afraid there is no progress in understanding the roles of FANCD2 on lipid metabolism yet. The last paragraph of "Introduction" and the last two sentences on page 9 should be rewritten.

We agree with the reviewer's comment that the molecular mechanism of FANCD2 in the lipid metabolism remains unclear. In addition to the limitation of this study in the original manuscript, the revised manuscript was modified according to the reviewer's suggestion (page 4 line 11, and page 10 line 19).

To address the reviewer's concern, we include new data on the effect of FANCC depletion in the FANCD2 localization to nLDs in the revised manuscript (Figure 4). FANCC is one of the FANCD proteins consisting of Fanconi anemia core complex. FANCC depletion prevents nuclear foci formation of FANCD2 upon MMC exposure, because Fanconi anemia core complex does not function in DNA damage response in the absence of FANCC (Figure S3C). Interestingly, our results demonstrate that the FANCC depletion by siRNA significantly suppresses the FANCD2 localization to nLDs (Figure 4F). These new data are important for understanding the possible roles of FANCD2 and other FANCD proteins in lipid metabolism. Discussion on this point is included in the main text (page 8 line 8), and possible model are shown in Figure S3D.

Specific points.

1. Figure 1A. The explanation of the "enrichment score" is insufficient. It is nice to show the equation to draw the score if it is helpful for the readers to figure out what the score represents. Or, p-values equivalent for enrichment score 0.67, 0.54, 0.24 should be indicated.

We sincerely thank the reviewer #1 for this crucial comment. To help understanding what the score represents, we included specific explanations about the enrichment score in materials and methods (page 13 line 27), and in Figure 1A legend (page 18 line 6).

2. The abbreviation PML should be spelled out at their first appearance.

According to the reviewer's comment, the text was modified (page 6 line 23).

3. Figure 5. Video data should be provided to show the photo bleach experiments.

We thank the reviewer #1 for this suggestion. We included video data showing fluorescence recovery after photobleaching as supplementary videos 3-5 (page 20 line 10, and page 22 line 6).

4. The legend for Figure 5 is poor. Experimental conditions should be described to help understanding to the readers. The left panel of Figure 5B may be divided into two separate graphs.

We highly appreciate this reviewer's comment, which must be a very important point. In the revised manuscript, we included detailed explanations of the experimental conditions in the legend of Figure 5 (page 20). Furthermore, two columns (bleached area and ROIs) were added to Figure 5A to help the reader's understanding. Accordingly, the fluorescent recovery of each ROI (a-d) was shown as four separate graphs in the Figure 5B.

5. *Summary, line 7. What do you mean for “lipid metabolic-related factors”. If you think of a particular enzyme(s) or pathway, please mention it rather than an ambiguous expression.*

We thank the reviewer #1 for this important comment. To avoid ambiguous expression, we modified the manuscript with a specific pathway (page 2 line 9).

6. *Introduction, line 8-12. Please site a relevant reference regarding ID complex formation and its importance on FA pathway.*

According to the reviewer's comment, we cited relevant papers indicating the importance of the ID complex in the FA pathway (page 3 line 9).

Reviewer #2

Major comments: Authors carefully conclude and present satisfactory data to support their claims.

1. *OPTIONAL: Proteomic analysis was performed on whole cell lysate. FANCD2 is a nuclear protein. Co-purification of FANCD2 in the nuclear fraction could give even more accurate proteomic results.*

We tried to identify unknown candidates for FANCD2 interacting partners, and eventually, we believed that unexpected factors have been identified by using total cell lysates. However, we agree with the reviewer #2 that using cell fractionation lysate in co-immunoprecipitation could lead to more specific results. Therefore, in the future project, we would like to perform another screening using cell lysate from the nuclear fraction to identify specific interactors of FANCD2 under the condition of oleic acid exposure.

2. *OPTIONAL: As the authors discussed in the last part of the manuscript, the question of whether the presence of FANCD2 on nLDs affects LDs' function is still to be answered. Experiments which could shed more light on that issue could be either i) proteomic analysis of co-purified FANCD2 partners under OA treatment - nuclear fraction. ii) alternatively, proteomics of isolated nLDs from nuclear fraction.*

We thank reviewer #2 for this important point. As already discussed above, to address the remaining issue in this study, we would like to focus on the more specific condition and try another screening in the future as suggested by reviewer #2.

Minor comments:

1. *To obtain fluorescent images, authors in M&M reveal the use of confocal microscopy. However, it is unclear if the images presented in the main figures are maximum intensity projections or single plain images. It is essential to also include in supplementary data one example of one dataset sequence of confocal images covering an entire volume of cells with DAPI and LDs staining. It will allow the differentiation of the nuclear LDs from the cytoplasmic ones.*

We thank the reviewer #2 for this crucial suggestion. All images in this study are single plan images, therefore the materials and methods in the revised manuscript is modified accordingly (page 15 line 2).

FANCD2 localization is totally same as nucleus. This is consistent with our and others previous data demonstrating that cellular FANCD2 localization almost represents the nucleus (Sakai et al, *FEBS Letters* 588, 3778-3785, 2014., Roques et al, *the EMBO Journal* 28, 2400-2413, 2009., Howlett et al, *Journal of Biological Chemistry* 284, 28935-28942, 2009., Hejna et al, *Journal of Biological Chemistry* 283, 9844-9851, 2008). Furthermore, our CLEM data (Figure 2G, H) demonstrates that LDs with FANCD2 obviously existed in the nucleoplasm and were not cytoplasmic LDs inside the invagination of the nuclear membrane. However, as the reviewer #2 points out, it is very important to distinguish between nuclear LD and cytoplasmic LD in this study. Therefore, to address the reviewer's concern, a dashed line indicating the location of FANCD2 (same as nucleus) were added to all images of the LDs, and we modified all figure legends in the revised manuscript accordingly.

2. *Authors in the discussion paragraph listed three reasons for the observed heterogeneity in the localization of FANCD2 to just a fraction of nLDs. However, I suggest discussing an additional factor, the LD's core content, which could also affect protein occupancy.*

We highly appreciate this valuable suggestion. This is related to the major comment (point 4) from reviewer #1. As the reviewer #2 kindly suggested, the hypothetical relationship between LD core lipids and surface proteins is now discussed as the 4th hypothesis in the revised manuscript (page 11 line 8).

3. In the lipid droplet and lipid metabolism field, the "FA" abbreviation is very often used to describe fatty acids. If possible, I propose to change the Fanconi anemia abbreviation.

As the reviewer pointed out, the abbreviation for Fanconi anemia (FA) may be confusing to some readers, so it is presented without the abbreviation (FA) and Fanconi anemia proteins are described as FANC proteins in the revised manuscript.

4. In the following sentence, it is not clear which "this conditions" are: "However, unlike FANCD2, FANCI exhibited no nLD localization under this condition " I propose to spell out the OA conditions.

According to the reviewer's thoughtful comments, the text was modified (page 7 line 31).

5. I propose not to introduce the ICL abbreviation. as it is mentioned only two times in the text.

The abbreviation for ICL were rephrased as suggested.

6. Not correct citation of the figure panel in the following sentence: "To test the hypothesis that FANCD2 functions in not only the DNA damage response but also fatty acid metabolism, we analyzed the subcellular localization of EGFP-FANCD2 (Fig S1D) under conditions of increased fatty acid levels."

We thank the reviewer #2 for this critical comment. The wrong citation was corrected (page 5 line 29).

Reviewer #3

Major comments

- The authors begin by highlighting 135 potential candidate FANCD2 interactors, but do not provide a full data set with information about the specific peptides observed, the number of times individual peptides were detected, or probability data. The details of biological and technical replicates analyzed in the experiment are unclear to this reader. The detailed proteomics data also should be deposited in an appropriate public database. Although it could be considered beyond the scope of the current manuscript, there is no follow up with characterization (i.e. co-IP validations or colocalization studies using immunofluorescence) for any of the candidate proteins from the screen.

We thank the reviewer #3 for this valuable suggestion. In the revised manuscript, we modified the material and methods (page 13 line 24), and included a new table regarding the detailed data set of 135 proteins list with protein score, number of peptide sequence matches, etc. (Supplementary Table S2). Regarding the deposit of our proteomics data in public databases, uploading was not possible due to unrecoverable data loss in some data, although we tried. We agree with the last part of reviewer's concern that follow-up analysis of FANCD2 binding partners is important. To address this point, we would like to try further analysis of each interaction in the future.

- The manuscript is limited to the analysis of one cancer cell line, U2OS, and it appears one CRISPR edited FANCD2 KO clone. It would be useful to know whether the localization of FANCD2 at nLDs is conserved in other systems other than U2OS cells, including normal non-malignant cells as well as other cancer cell lines. The methods suggest that multiple KO clones were generated and the paper would be strengthened by data confirming that the results are consistent across multiple clones (although importantly fatty acid phenotypes are largely rescued by reintroducing FANCD2 into the KO cells).

This is related to the major comment (point 1) from reviewer #1. U2OS is a cell line derived from human osteosarcoma. Human hepatoma cell line, such as Huh-7 and HepG2, are used to study lipid metabolism and lipid droplet formation. However, our study focuses not only on lipid metabolism but also on DNA damage response. U2OS cell line has functional p53 and is suitable to analyze DNA damage response and DNA repair. Many papers are published using U2OS in research of DNA damage response. Importantly, this cell line has been widely used to analyze nuclear lipid droplet formation (Ohsaki et al, *J Cell Biol* 212, 29-38 (2016). Wong et al, *Cell Death Discovery* 4, 109 (2018). Soltysik et al, *Nat Commun* 10, 473 (2019). Lee et al, *Life Sci Alliance* 3, 8, (2020). Ivanovska et al, *J Cell Biol* 222, 8 (2023)). From these points, we believe that U2OS is advantageous for our research. To address the reviewer's concern, materials and methods in the revised manuscript was modified (page 12, line 6). In the future projects, we would like to examine the FANCD2 dynamics in response to oleic acid exposure in other types of cell lines including the normal non-malignant cells.

We generated some FANCD2^{KO} clones and the wild-type FANCD2-expressing cells from the KO clones. To verify the phenotype of FANCD2 deficiency, these cells were examined mitomycin C (MMC) sensitivity. However, only one KO clone was fully rescued from MMC sensitivity by ectopic expression of wild-type FANCD2, which clone used in this study. This suggests that off-target effects from the gene disruption may exist in the other KO clones. To demonstrate that the FANCD2^{KO} clone in this study is a bona fide, a complementation test for MMC sensitivity is included in Figure S1D.

- The localization of FANCD2 to nLDs occurs without FANCI, is not stimulated by MMC, and doesn't require FANCD2 ubiquitination. These are interesting data that distinguish this mechanism from the canonical DDR role of FANCD2. The authors cite work showing that metabolic alterations occur in many FA patients (presumably independent of complementation group) and that lipid metabolism is reported to be altered in FANCA mutant cells. It would be worthwhile to test whether the localization of FANCD2 to nLD requires any core complex factors or is entirely independent of them. The authors also provide some data suggesting a potential competition between FANCD2 and FANCI at lipid droplets, but overall it is unclear from the study how the proposed role for FANCD2 fits into the FA pathway overall. As noted by the authors as a study limitation, the work also does not address the downstream mechanisms by which FANCD2 modulates lipid metabolism.

We thank the reviewer #3 for this crucial suggestion. In the revised manuscript, we include new data on the effect of FANCC depletion in the FANCD2 localization to nLDs (Figure 4). FANCC is one of the FANCD proteins consisting of Fanconi anemia core complex. FANCC depletion prevents nuclear foci formation of FANCD2 upon MMC exposure, because Fanconi anemia core complex does not function in DNA damage response in the absence of FANCC (Figure S3C). Interestingly, our results demonstrate that the FANCC depletion by siRNA significantly suppresses the FANCD2 localization to nLDs (Figure 4F). The mechanism by which FANCD proteins regulate lipid metabolism remains unclear. However, these new data are important for understanding the possible roles of FANCD proteins in lipid metabolism. Discussion on this point is included in the main text (page 8 line 8), and possible model are shown in Figure S3D.

Minor comments

- Given the low percentage of nLD with FANCD2 present (Fig. 2D/E), it would be helpful if the total number lipid droplets observed per nuclei in untreated and OA treated U2OS cells was also reported.

We thank the reviewer #3 for raising this crucial point. The data showing the total number of nLDs per nucleus is helpful in understanding this study. Therefore, we included a new data of nLD frequency in the condition with and without OA exposure (page 6 line 4, and Figure S1E).

- Figure 2A/B: what is the percentage observed between retained FANCD2 localization versus loss of FANCD2 localization over time? Quantitative data and discussion about the turnover of FANCD2 at these droplets would be beneficial. As suggested in the manuscript, it would be worthwhile to explore the possibility of FANCI localizing at these regions as FANCD2 is lost.

We agree with the reviewer #3 that such quantitative data for the dynamics of FANCD2 upon OA exposure is important. As already discussed above, however, the localization of FANCD2 to nLD is low frequency, and even if it could be detected, it would be difficult to obtain sufficient data for statistical analysis. On the other hand, it is beneficial to discuss the turnover of FANCD2/nLD. Therefore, we cited the following paper on the cell cycle and lipid droplet biosynthesis (Cruz et al, *Molecular and Cellular Biology*, e00374-18, 2019), and discussed this point in the revised manuscript (page 11 page 24).

We agree with the point that it would be valuable to elucidate the localization balance between FANCI and FANCD2 on nLDs. As already discussed in the original manuscript, FANCI localization on nLDs cannot be detected without depletion of FANCD2 expression (Figure 4). Even if FANCD2 disappeared from nLDs (FANCD2 is still present in the cell), it would be difficult to detect the FANCI localization on nLDs.

Original submission

First decision letter

MS ID#: jcs.264430

MS Title: Dynamics of Fanconi anemia protein D2 in association with nuclear lipid droplet formation

Authors: Tomoya Hotani; Motonari Goto; Yukie Otsuki; Shun Matsuda; Nobuhiro Wada; Masakazu Shinohara; Tomonari Matsuda; Masayuki Yokoi; Kaoru Sugasawa; Yuki Ohsaki; Wataru Sakai
Article Type: Review Commons Transfer

Dear Dr Sakai,

Thank you for sending your revised manuscript, initially assessed through the Review Commons mechanism, to JCS for further consideration. I am happy to tell you that your manuscript has been accepted for publication in Journal of Cell Science, pending standard publication integrity checks. It was accepted on 12 September 2025. Where referee reports on this version are available, they are appended below.

Thank you for sending your manuscript to Journal of Cell Science through Review Commons.